# Addition of Phosphorous and IL6 to m-EASIX Score Improves Detection of ICANS and CRS, as Well as CRS Progression

**DOI:** 10.3390/cancers17060918

**Published:** 2025-03-07

**Authors:** Kenneth Barker, Tom Marco, Muhammad Husnain, Emmanuel Katsanis

**Affiliations:** 1Department of Medicine, University of Arizona, Tucson, AZ 85724, USA; 2Department of Hematology and Oncology, University of Arizona Cancer Center, Tucson, AZ 85719, USA; 3Departments of Immunobiology, Medicine, and Pathology, University of Arizona, Tucson, AZ 85724, USA; 4Department of Pediatrics, University of Arizona, Tucson, AZ 85724, USA

**Keywords:** chimeric antigen receptor T cells, CAR-T, cytokine release syndrome, CRS, immune effector cell-associated neurotoxicity, ICANS, biomarkers, hypophosphatemia, modified endothelial stress activation index, m-EASIX

## Abstract

CAR-T therapy has revolutionized the treatment of B-cell malignancies. However, its increasing use has been accompanied by a rise in CAR-T-related toxicities and associated morbidity. Currently, no widely adopted clinical prediction models exist for identifying patients at risk for these toxicities. Early prediction could enable timely interventions to mitigate adverse effects. Phosphorus, an inexpensive and routinely measured lab value, has recently been shown to decline before the onset of CAR-T toxicity. Additionally, interleukins such as IL-6, though less commonly included in routine lab monitoring, have also been linked to CAR-T-related toxicity. This study builds upon existing predictive models of CAR-T toxicities by incorporating these biomarkers to enhance early detection and intervention, ultimately aiming to reduce CAR-T-associated morbidity.

## 1. Introduction

Chimeric antigen receptor (CAR) T cell therapy has fundamentally altered how we approach the treatment of B-cell malignancies. However, its wider adoption has brought about a rise in CAR-T-related adverse effects, prominently seen in cytokine release syndrome (CRS) and immune effector cell-associated neurotoxicity syndrome (ICANS) [1,2]. CRS and ICANS are common and potential life-threatening complications of CAR-T therapy. Early recognition of these conditions is vital to preventing CAR-T-related morbidity and mortality [2,3]. Numerous biomarkers, including C-reactive protein (CRP), ferritin, IL-6, IL-10, and IFN-γ, have been associated with both syndromes [4,5,6]. However, there is currently no standardized model routinely employed in clinical practice to predict either syndrome.

The pathogenesis of CRS and ICANS is thought to be dependent on endothelial cell dysfunction, prompting the extension of the endothelial activation and stress index (EASIX) score to predict the probability of developing severe grades of CRS and ICANS following CAR-T infusion [7,8]. The EASIX score is generated based on the following formula: log2((Lactate dehydrogenase (LDH) [U/L] × creatinine [mg/dL])/(platelets) [10^9^ cells/L]). It has been shown to be a useful surrogate marker for inflammatory-induced endothelial dysfunction and activation [9,10]. In the 2021 study by Pennisi et al. [11], a modified version of the EASIX (m-EASIX) was generated based on the following formula: log2((CRP [mg/dL] × LDH [U/L])/(platelets [10^9^ cells/L])). It was demonstrated to be able to predict severe grades of CRS and ICANS. The m-EASIX excluded creatinine, as no association between creatinine and CRS or ICANS was found, and instead replaced creatinine with CRP, another known biomarker associated with CRS and ICANS [12,13]. Another biomarker associated with endothelial dysfunction not yet studied for use in conjunction with the m-EASIX score is the pro-inflammatory cytokine IL6 [14]. It is a known contributor to vascular endothelium instability during CRS and ICANS, with rising peak levels linked to the development of both syndromes [15,16].

An emerging biomarker of interest associated with the development of both CRS and ICANS is phosphorous. The study by Tang et al. first demonstrated in vitro that increased consumption of phosphorous was associated with increased levels of CAR-T immune system activation to meet higher levels of metabolic demand induced by overaction of the immune system [17]. Since then, several studies, including our own, have revealed that hypophosphatemia often precedes the onset of CRS or ICANS, and the occurrence of rapidly dropping levels of phosphorous after CAR-T infusion provide a potential a biomarker indicating the impending development of CRS or ICANS [18,19,20].

Our current study aimed to improve the early recognition of CRS and ICANS by incorporating phosphorous and IL6, together and separately, into the m-EASIX score.

## 2. Methods

This was a retrospective study using patients undergoing CAR-T treatment for non-Hodgkin lymphoma (NHL) at the University of Arizona Cancer Center. The patients received CAR-T products of either Tisagenlecleucel or Abxicabtagene ciloleucel. Standard lymphodepleting therapy was administered prior to infusion as per a previously described protocol [21,22]. All patients were screened and negative for active infection as per institutional protocol. Eastern Cooperative Oncology Group (ECOG) guidelines were used to measure functional status prior to CAR-T infusion. The international prognostic index (IPI) score was recorded for each patient according to IPI guidelines. The day of symptom onset for CRS, ICANS, and/or progression of CRS to grade ≥ 2 was recorded and graded following ASTCT guidelines. The median day of onset for each of the investigated CAR-T toxicities was recorded. IL-6 inhibitors, IL-1 inhibitors, and steroids were administered following the institutional protocol, independent of the investigated inflammatory marker values, including the IL-6 values. Collection of data was performed with approval from the University of Arizona Institutional review board and in accordance with the declaration of Helinski.

Phosphorous, platelets, CRP, IL-6, and LDH were recorded daily up until 7 days before and 14 days after CAR-T infusion. IL-6 values were recorded on the day of CAR-T infusion and every 3 days after due to the lab value needing to be sent out to be measured outside our institution. Serum IL6 < 2 pg/mL and CRP < 3 mg/dL were unable to be quantified due to limits of laboratory detection and were rounded up to 2 pg/mL and 3 mg/dL, respectively, for analysis. Median values for each measured day were recorded and plotted over time. These values were then used to generate the following variations in the m-EASIX score, as originally described in 2021 by Pennisi et al. [11]:

m-EASIX: Log2((CRP [mg/dL] × LDH [U/L])/(Platelets [10^9^ cells/L]))

P-m-EASIX: Log2((CRP [mg/dL] × LDH [U/L])/(Platelets [10^9^ cells/L] × Phosphorous^3^ [mg/dL]))

IL6-m-EASIX: Log2((CRP [mg/dL] × LDH [U/L] × IL6 [pg/mL])/(Platelets [10^9^ cells/L]))

P-IL6-m-EASIX: Log2((CRP × LDH × IL-6 (pg/mL))/(Platelets [10^9^ cells/L] × Phosphorous^3^ [mg/dL]))

All scores underwent log-transformation using base 2, as described in both the original EASIX report and m-EASIX score, to decrease skewness. The values of phosphorous were raised to the 3rd power to help ensure that the relative differences in the phosphorous had a proportional consistency to the differences in other values in the investigated scores. The scores were calculated for potential clinically actionable timepoints of day of infusion (day +0), day +1 post infusion, day +2 post infusion, and day +3 post infusion. Scores including IL6 were unable to be calculated on day +1 and day +2 due to an insufficient number of measurements. All calculated scores were used to generate ROC curves for the occurrence of CRS, ICANS, and the progression of CRS to grade ≥ 2, with statistical significance calculated using the Wilson/Brown method. Univariable logistic regression was performed on all variations in scores at the measured time points to determine OR and CIs.

All variables in the scores underwent individual univariable logistic regression to assess the significance of contribution of each component of the measured scores for the occurrence of CRS, ICANS, and the progression of CRS to grade ≥ 2. All statistical analyses and figures were generated with Prism V10.2.2.

## 3. Results

### 3.1. Patient Characteristics

A total of 42 patients (24 male and 18 female) with NHL, ranging from 42 to 86 years of age, presenting to the University of Arizona Cancer Center from January 2019 to October 2023 successfully underwent CAR-T treatment (Table 1). Thirty-three patients (79.0%) received tisagenlecleucel, and nine patients (21%) received abxicabtagene ciloleucel. In terms of lymphodepletion, 33 (79%) received fludarabine/cyclophosphamide and 9 (21%) received bendamustine (due to a national shortage of fludarabine at the time of lymphodepletion) [22]. Prior to CAR-T infusion, 38 patients had an ECOG score of 0–1 and 4 patients had an ECOG score of 2–3. with regard to prognosis and overall progression-free survival based on international prognostic index (IPI) scoring, 16 patients (38%) were low/low–intermediate risk and 26 (68%) were intermediate/intermediate–high risk. Ten (24%) patients had received a prior stem cell transplant. We did not find an association with any of the described patient characteristics and the development of CRS, CRS progression to grade ≥ 2, and ICANS.

CRS, CRS progression to grade ≥ 2, and ICANS occurred in 32 (76%), 18 (43%), and 12 (29%) patients, respectively. An analysis of severe CRS and ICANS was unable to be performed due to only one patient developing severe CRS (grade 3) and two developing severe ICANS (grade 4). The median time of onset was +2, +4, and +5 days following CAR-T infusion for CRS, CRS progression to grade ≥ 2, and ICANS, respectively.

### 3.2. Association of the Individual Variables in the Calculated Scores with CRS, CRS Progression to grade ≥ 2, and ICANS

Platelet counts were significantly associated with the occurrence of CRS, CRS progression to grade ≥ 2, and ICANS on day +0, +1, +2, and +3 (Table 2). The OR for developing ICANS or CRS progression slightly decreased over time from day +0 (CRS progression OR 0.880; *p* = 0.023, ICANS OR 0.862; *p* = 0.025) to day +3 (CRS progression OR 0.819; *p* = 0.0076, ICANS OR 0.746; *p* = 0.0038) for increasing platelet levels. The OR associated with CRS and platelets remained similar for days +0 (OR 0.796; *p* = 0.0031) to +3 (OR 0.805; *p* = 0.0070). Phosphorous levels became significantly associated with the occurrence of ICANS on day +1 (OR 0.921; *p* = 0.0024) and remained significant through day +3 (OR 0.803; *p* = 0.0003). This was in congruence with the decreasing median phosphorous levels over time, starting on day +1 from infusion and reaching nadir levels on day 5. Hypophosphatemia was significantly associated with the occurrence of both CRS and CRS progression starting on day +1 (CRS OR 0.874; *p* = 0.032, CRS progression OR 0.827; *p* = 0.010). The significant association carried on through day +3 (CRS OR 0.803; *p* = <0.0001, CRS progression OR 0.926; *p* = 0.047). CRP was significantly associated with ICANS on day +2 (OR 1.56; *p* = 0.025) and day +3 (OR 1.42 *p* = 0.0037).

CRP was also significantly associated with CRS and CRS progression, peaking on day +3 (CRS OR 4.21; *p* = 0.025, CRS progression OR 1.43; *p* = 0.016). Serum IL6 levels on day +0 were not significantly associated with CRS, CRS progression, or ICANS. Serum IL-6 data was too limited on day +1 and day +2 to analyze. On day +3, the IL6 levels were significantly associated with all measured CART toxicities (CRS OR 1.18; *p* = 0.0009, CRS progression OR 1.52; *p* = 0.016, and ICANS OR 1.39: *p* = 0.0034). The LDH values were not significantly associated with the studied CAR-T toxicities at any time point.

### 3.3. P-m-EASIX Score Demonstrated the Greatest Discriminatory Capabilities for ICANS and CRS

P-m-EASIX showed the highest discriminatory capabilities on ROC analysis and the highest OR on univariate logistic regression for the prediction of ICANS and CRS. P-m-EASIX was significantly associated with the development of ICANS on day +0 (OR 1.43; *p* = 0.047), and on day +1 (OR 2.23; *p* = 0.0096), it had the largest delta day-to-day increase, where it remained similar on day +2 (OR 2.24; *p* = 0.0096) and day +3 (OR 2.22; *p* = <0.0001) (Figure 1).

P-m-EASIX’s performance on ROC analysis in predicting ICANS was the highest out of the measured scores: day +0 (AUC 78.9%; *p* = 0.0073), day +1 (AUC 89.6%; *p* = 0.0090), day +2 (AUC 93.0%; *p* = 0.0001), and day +3 (AUC 91.5%; *p* =< 0.0001) (Figure 2).

P-m-EASIX’s performance for the prediction of CRS and CRS progression on day +0 and day +1 was similar to m-EASIX, in that there was no significant association upon univariate logistic analysis or ROC analysis until day +2 (CRS OR 1.79; *p* = 0.014, AUC 82.3%; *p* = 0.0030, CRS progression OR 1.79; *p* = 0.014, AUC 78.1%, *p* = 0.0045). Out of all the calculated scores, the P-m-EASIX score showed the third highest peak discriminatory function for CRS progression on day +3 (OR 1.51; *p* = 0.00031, AUC 84.7%; *p* = 0.0001) compared to the m-EASIX score (OR 1.59; *p* = 0.0015, AUC:79.9%; *p* = 0.00070). The P-m-EASIX score showed the highest predictive capabilities for CRS on day +3 (OR 2.21; *p* = 0.0014, AUC 92.0% *p* =< 0.0001).

### 3.4. IL6-m-EASIX Had the Highest Initial Discriminatory Capabilities for CRS Progression to grade ≥ 2

Analysis of the IL6-m-EASIX score was limited on day +1 and day +2 due to the limited availability of IL6 serum levels. IL6-m-EASIX was similar to the P-m-EASIX score and m-EASIX score, in that it was significantly associated with the development of ICANS on day +0 (OR 1.31; *p* = 0.01, AUC 83.8%; *p* = 0.039). The combination of phosphate and IL6 into the P-IL6-m-EASIX score showed the highest discriminatory function for ICANS on day +0 (OR 1.85; *p* = 0.041, AUC 86.8%; *p* = 0.023) compared to m-EASIX on day +0 (OR 1.49; *p* = 0.012, AUC 77.9%; *p* = 0.0097). The addition of IL6 levels alone in IL6-m-EASIX showed the largest increase in AUC for ROC analysis compared to the other scores in the discriminatory function for predicting CRS progression to grade ≥ 2 on day +3 (OR 1.57; *p* = 0.031, AUC 89.7%; *p* = 0.0040) and compared to m-EASIX (OR 1.59; *p* = 0.0015, AUC 79.9%; *p* = 0.0011).

## 4. Discussion

### 4.1. Phosphorous: P-m-EASIX

The addition of phosphorous successfully augmented the predictive capabilities of the m-EASIX score for ICANS, CRS, and CRS progression. The enhanced predictive capabilities for ICANS over time for the P-m-EASIX score followed the timing of the drop in phosphorous seen during the development of ICANS in both our study and others, with levels beginning to drop as early as day +1 (Figure 3) [17,19,23,24]. This was reflected in the largest increase in the discriminatory capabilities of P-m-EASIX for predicting ICANS onset from day +0 to day +1. Peak function was achieved on day +2, a fact which, if used clinically, could guide treatment decisions to mitigate complications before the median onset of ICANS on day +5. Similarly to the m-EASIX score, we found P-m-EASIX to have limited utility in predicting CRS on day +0 and day +1 before the median onset of first CRS symptoms on day +2. However, P-m-EASIX showed promise in predicting CRS progression at an increased discriminatory capacity compared to m-EASIX alone by day +3, which would be a clinically actionable time point considering that the median onset of CRS progression was on day +4.

Phosphorous levels were additionally independently associated with the occurrence of ICANS, CRS, and CRS progression. Another approach to the manner in which phosphorous is incorporated into the m-EASIX score would be using the absolute change from baseline levels on day +0, based on the study by Yoshida et al. [23], showing the absolute change to be more predictive of CAR-T toxicity. Phosphorus levels remained stable during the lymphodepletion period before CAR-T infusion, suggesting that the observed decline in phosphorus levels was a direct effect of the infused CAR-T cells (Figure 3). The mechanism behind the drop seen in phosphate levels with both CRS and ICANS appears to be multifactorial, as previous studies have shown that CAR-T-induced overaction in the immune system can increase both metabolic systemic ATP consumption and induced renal phosphate wasting (Figure 4). Regardless of the mechanism, the addition of phosphate to the m-EASIX score provides an inexpensive and easy method to enhance its predictive capabilities for the early detection of ICANS, CRS, or CRS progression.

### 4.2. IL6: IL6-m-EASIX

The addition of IL6 to the m-EASIX score improved the predictive capabilities for CRS and ICANS, though it was slightly less effective than P-m-EASIX. However, IL6 addition showed the greatest enhancement for predicting CRS progression on day +3, indicating its utility in identifying patients prone to CRS progression. Due to limited IL6 data in our cohort, it is challenging to draw generalized conclusions about IL6-m-EASIX.

IL6 was independently associated with all three measured occurrences. Drawbacks of using IL6 levels are the limited availability and longer time required to obtain serum levels compared to other EASIX components, which limit its clinical utility. Analyzing larger cohorts of CAR-T-associated CRS and ICANS with serum IL6 data could provide further insights into their predictive utility.

### 4.3. The Role of Thrombocytopenia in CRS and ICANS

Platelet levels were unique in our analysis of components of the EASIX score, in that they were the only value we independently found to be associated with the occurrence of ICANS, CRS, and CRS progression on day +0. This suggests that, on day +0, platelets had the greatest contribution to all variations in the m-EASIX scores, enhancing the predictive capabilities for the studied CAR-T toxicities. This is in line with previous studies, showing that the presence of thrombocytopenia prior to infusion is associated with the occurrence of CRS and ICANS [24,25,26]. The presence of thrombocytopenia is thought to result in an increased permeability of both the peripheral and brain vascular endothelial membranes through multiple mechanisms, including the decreased production of ang-1, a known stabilizer of the vascular membrane, lower levels of which have been associated with an increasing severity of CRS and ICANS [27,28,29]. Platelet levels are ubiquitously available as routine lab measurements and remain an important component in all variations in the EASIX scores investigated.

### 4.4. m-EASIX 

Similarly to the study by Acosta-Medina et al. [30], we did not find any statistically significant cutoff with the m-EASIX score on day +0 for the occurrence of CRS or CRS progression upon ROC analysis, even with the addition of phosphate and/or IL6.

Also similarly to previous studies, m-EASIX was able to significantly predict the occurrence of ICANS on day +0, and on day +0, the addition of IL6 and phosphorous did little to augment m-EASIX’s predictive capabilities for ICANS. The AUC increased for CRS, CRS progression, and ICANS, as expected in the days post CAR-T infusion, which coincided with the fluctuations in CRP and platelet levels seen in Figure 3. We did not find an association with LDH on day 0 or day 3 with CRS, CRS progression, or ICANS upon univariate logistical regression. The m-EASIX score remained, in our study, a reliable and inexpensive calculation of CAR-T toxicity using routinely available lab data.

## 5. Conclusions

The inclusion of phosphorous and, when available, IL6 enhanced the m-EASIX score’s discriminatory capabilities for predicting CRS, CRS progression, and ICANS. Our sample size limited the analysis of these scores’ ability to predict progression to severe CRS and ICANS. These variations in the m-EASIX score aim to help predict CAR-T-related toxicities and guide clinical decision making by identifying patients who may benefit from early interventions, such as steroids, IL-6 inhibitors, and IL-1 inhibitors, to prevent CAR-T toxicity onset and progression. Larger-scale studies to validate improved predictive capabilities with the addition of phosphorous and IL6 could help guide the early management of complications seen in both ICANS and CRS following CAR-T therapy.

## Figures and Tables

**Figure 1 cancers-17-00918-f001:**
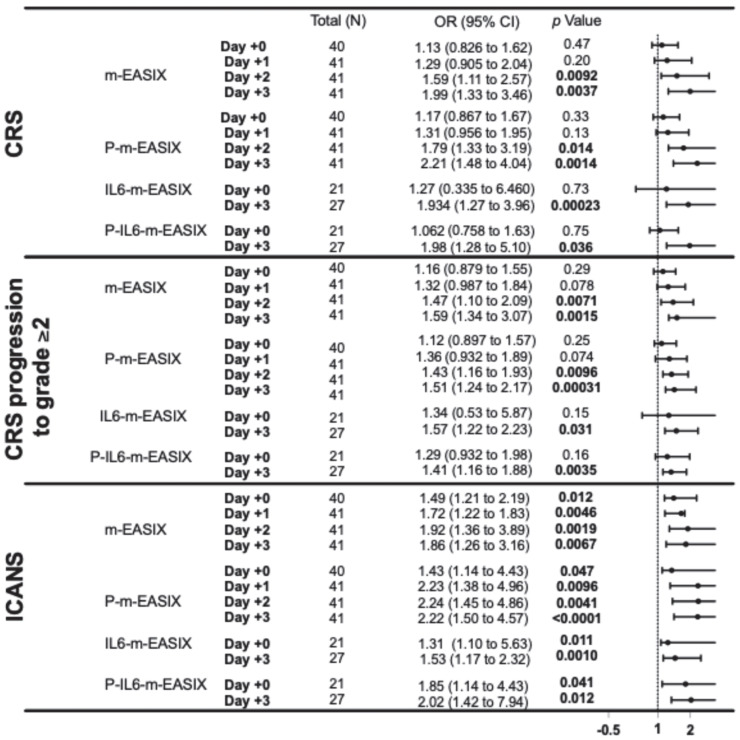
This forest plot describes the ORs and 95% CI for all variations in the m-EASIX score, including the original for the occurrence of CRS, CRS progression to grade ≥ 2, and ICANS at all measured time points, relative to the day of CAR-T infusion (day +0). Significance and OR were generated through univariate logistic regression.

**Figure 2 cancers-17-00918-f002:**
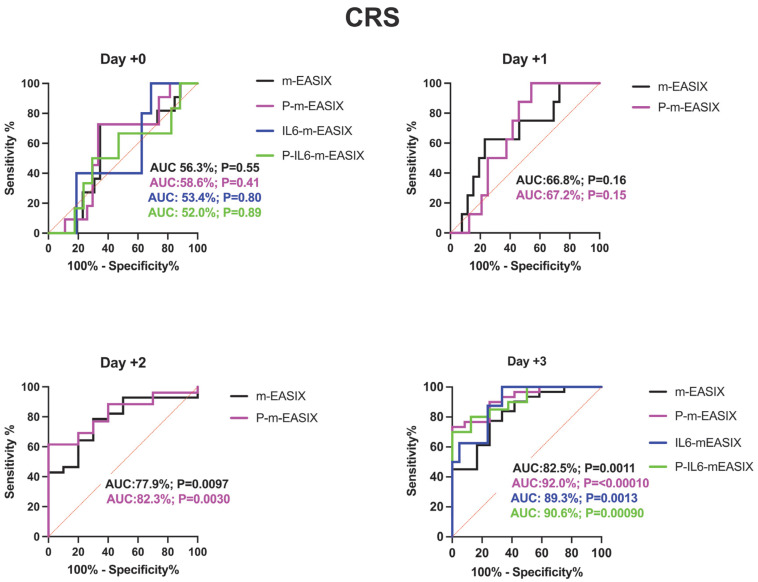
ROC analysis for the occurrence of CRS, ICANS, and progression of CRS to grade ≥ 2. Day +1 and day +2 present only the m-EASIX and P-m-EASIX scores, as both days had insufficient IL-6 data to generate ROCs for IL6-containing equations. Statistical significance was calculated using the Wilson/Brown method.

**Figure 3 cancers-17-00918-f003:**
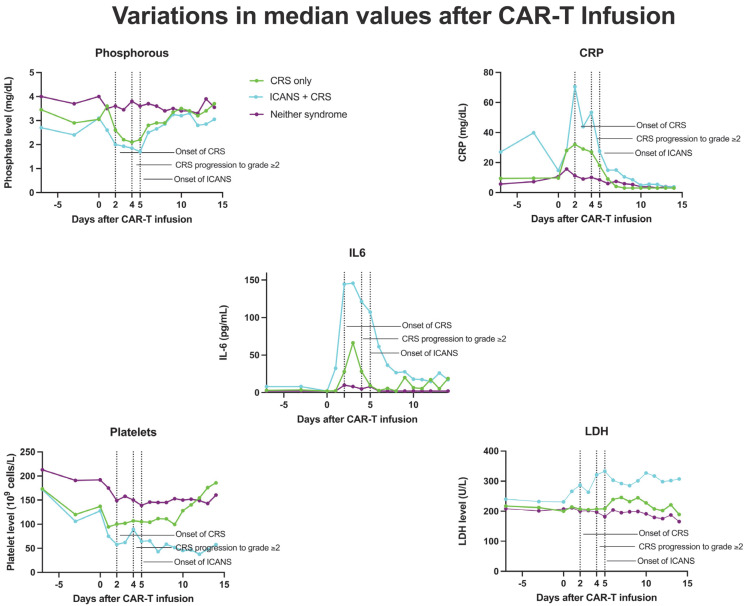
Median values of CRP, phosphorous, IL-6, platelets, and LDH in patients who developed CRS only, ICANS+CRS, or neither syndrome following CAR-T infusion, from day 7 to day 14.

**Figure 4 cancers-17-00918-f004:**
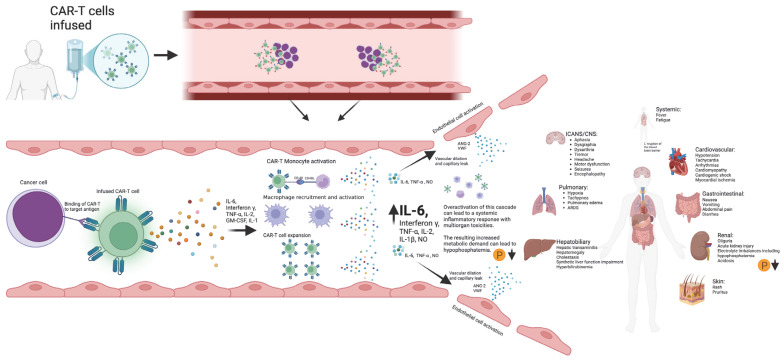
Schematic diagram outlining the development of CAR-T toxicities from CAR-T antigen recognition to subsequent cytokine storm and multiorgan toxicities. Created in BioRender. Barker, K. found at https://BioRender.com/i96l898 (accessed 24 February 2025).

**Table 1 cancers-17-00918-t001:** Patient characteristics.

Patient Characteristics	
Age, median (range)	
Sex (n)	
Female	26
Male	16
Non-Hodgkin’s lymphoma subtype (n)	
Diffuse large B-cell lymphoma (DLBCL)	40
Follicular cell lymphoma	2
CAR-T cell product	
Tisagenlecleucel	33
Abxicabtagene ciloleucel	9
ECOG pre infusion (n)	
0–1	38
2–3	4
4–5	0
IPI pre infusion (n)	
0–2 low/low–intermediate risk	16
3–5 intermediate/intermediate–high risk	26
≥1 site of extranodal disease	
Yes	24
No	18
Lymphodepleting regimen	
Fludarabine/cyclophosphamide	33
Bendamustine	9
Previous stem cell transplant	
Yes	10
No	32

ECOG, Eastern Cooperative Oncology Group; IPI, international prognostic index.

**Table 2 cancers-17-00918-t002:** Analysis of individual components of each score.

	Median (Range)	*N*	*Odds Ratio*	*95% CI*	*p Value*
**CRS**					
*Day +0*					
Platelets *•*	11.9 (1.10–22.9)	42	0.796	0.650 to 0.912	**0.00031**
Phosphorous °	33.5 (25.0- 46.0)	42	0.720	0.273 to 1.85	0.50
IL-6 *	2.00 (2.00–15.8)	21	2.02	0.953 to 38.7	0.43
CRP *	0.954 (0.301–2.08)	40	1.43	0.178 to 2.88	0.62
LDH *	2.32 (2.100–2.84)	40	1.13	0.743 to 1.732	0.78
*Day +1*					
Platelets *•*	11.5 (1.00–20.2)	42	0.832	0.697 to 0.950	**0.016**
Phosphorous °	30.0 (19.0–39.0)	42	0.874	0.759 to 0.978	**0.032**
CRP *	1.24 (0.478–2.22)	42	1.39	0.925 to 2.32	0.28
LDH *	2.37 (2.10 to 2.91)	42	1.17	0.943 to 1.45	0.38
*Day 2*					
Platelets •	10.1 (1.00–17.6)	42	0.797	0.645 to 0.924	**0.0014**
Phosphorous °	22.0 (16.0 to 36.0)	42	0.730	0.556 to 0.873	**0.0048**
CRP *	1.72 (0.477 to 2.22)	42	1.93	1.07 to 5.32	**0.049**
LDH *	2.32 (2.09 to 2.84)	42	1.04	0.832 to 1.43	0.63
*Day 3*					
Platelets *•*	10.4 (2.1–16.8)	42	0.805	0.634 to 0.949	**0.0070**
Phosphorous °	22.2 (14.0–35.0)	42	0.728	0.546 to 0.862	**<0.0001**
IL-6 *	56.0 (2.00–637.3)	27	1.18	1.02 to 1.49	**0.0009**
CRP *	1.57 (0.826–2.4)	42	4.21	1.25 to 6.34	**0.0246**
LDH *	2.34 (2.14–2.52)	42	1.32	0.992 to 1.58	0.248
*ICANS*					
*Day 0*					
Platelets •	10.1 (1.20–11.1)	42	0.862	0.753 to 0.972	**0.025**
Phosphorous °	3.10 (2.50–4.80)	42	0.946	0.839 to 1.05	0.34
IL-6 *	2.00 (2.00–9.60)	21	1.06	0.801 to 1.36	0.62
CRP *	1.07 (0.477–2.08)	40	0.991	0.966 to 1.01	0.39
LDH *	2.37 (2.09–2.84)	40	1.01	0.998 to 1.09	0.12
*Day 1*					
Platelets *•*	8.9 (1.00 to 16.4)	42	0.821	0.698 to 0.931	**0.0024**
Phosphorous °	28.5 (20.0 to 34.0)	42	0.921	0.784 to 0.979	**0.0001**
CRP *	1.42 (0.672 to 2.22)	42	1.34	0.954 to 1.49	0.157
LDH *	2.42 (2.21–2.91)	42	1.73	0.869 to 2.85	0.34
*Day 2*					
Platelets *•*	6.45 (1.00 to 12.4)	42	0.781	0.641 to 0.913	**0.0009**
Phosphorous °	22.0 (19.0 to 29.0)	42	0.886	0.723 to 0.931	**0.0048**
CRP *	1.802 (0.897 to 2.21)	42	1.56	1.19 to 1.83	**0.025**
LDH *	2.46 (2.14–2.84)	42	1.435	0.812 to 2.17	0.39
*Day 3*					
Platelets *•*	6.55 (1.20- 11.1)	42	0.746	0.593 to 0.888	**0.0038**
Phosphorous °	19.5 (14.00–30.0)	42	0.872	0.748 to 0.976	**0.0036**
IL-6 *	146 (2.00–637.3)	27	1.39	1.09 to 1.71	**0.0034**
CRP *	1.73 (0.823- 2.34)	42	1.42	1.31 to 1.57	**0.0037**
LDH *	2.37 (2.16–2.43)	42	1.05	0.932 to 1.13	0.16
**Progression of CRS to grade ≥ 2**					
*Day 0*					
Platelets *•*	11.4 (1.10–24.5)	42	0.880	0.779 to 0.974	**0.023**
Phosphorous °	31.0 (25.0–46.0)	42	0.951	0.857 to 1.05	0.32
IL-6 *	2.00 (2.0–16.8)	21	1.31	0.991 to 2.17	0.14
CRP *	0.930 (0.301–2.082)	40	1.23	0.867 to 1.41	0.77
LDH *	2.32 (2.10–2.84)	40	1.56	0.836 to 2.26	0.47
*Day 1*					
Platelets *•*	10.8 (1.50 to 16.4)	42	0.827	0.702 to 0.942	**0.010**
Phosphorous °	28.0 (21.0–38.0)	42	0.926	0.823 to 1.02	0.15
CRP *	1.24 (0.478–2.22)	42	1.10	0.925 to 1.41	0.28
LDH *	2.38 (2.21 to 2.91)	40	1.38	0.773 to 1.79	0.38
*Day 2*					
Platelets *•*	9.70 (1.30 to 16.1)	42	0.796	0.658 to 0.923	**0.0074**
Phosphorous °	23.0 (20.0–35.0)	42	0.878	0.752 to 0.964	**0.010**
CRP *	1.26 (0.491 to 2.22)	40	1.34	0.845 to 1.687	0.32
LDH *	2.34 (2.14 to 2.84)	40	1.49	0.743 to 2.38	0.49
*Day 3*					
Platelets *•*	8.60 (1.20–13.2)	42	0.819	0.659 to 0.935	**0.0076**
Phosphorous °	20.0 (14.0–35.0)	42	0.862	0.764 to 0.943	**0.0040**
IL-6 *	111 (2.00–637.5)	27	1.52	1.26 to 1.78	**0.016**
CRP *	1.74 (.826–2.37)	42	1.43	1.11 to 1.89	**0.016**
LDH *	2.37 (2.16–2.51)	42	1.51	0.993 to 2.15	0.13

* A log-transformation was performed for CRP, IL6, and LDH values. ° Phosphorous values were scaled up by a factor of 10. • Platelets were scaled down by a factor of 10. Significance and odds ratios determined through univariate logistical regression. Significant values are shown in bold.

## Data Availability

The original contributions presented in this study are included in the article. Further inquiries can be directed to the corresponding author.

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
