# Peer review of "Addition of Phosphorous and IL6 to m-EASIX Score Improves Detection of ICANS and CRS, as Well as CRS Progression"

_cancers, 2025, doi:10.3390/cancers17060918_

Round 1
Reviewer 1 Report
Comments and Suggestions for Authors
The article titled 'The addition of phosphorous and IL6 to the m-EASIX score 2 improves detection of ICANS, CRS, and the progression of 3 CRS' by Barker et al is interesting. The article can be improved with the below comments.
- Given that the article heavily focuses on phosphorus levels and their correlation with other biomarkers such as CRP and IL-6, could the authors comment on the potential fluctuations in phosphorus levels during lymphodepletion in the patients?
- Based on the hypothesis, It is interesting to know that the levels of phosphorus can be corelated with CRS and its potential use as a possible biomarker.
- What are possible pathways that are involved in this, did the authors look into the mechanism by performing NGS on the samples ?
- Is there a direct co-relation of phosphorus and CAR-T therapy ?
- The article primarily focuses on phosphorus levels in grade-2 Cytokine Release Syndrome (CRS). Did the authors also investigate other grades of CRS? If so, can any conclusions be drawn from the available data ?
- Did the patients undergo any other treatment when IL-6 expression was higher such as the use of Tocilizumab or other IL-6 inhibitors for treatment.
- Can the authors include a schematic depiction of summary of the article.
Author Response
Comment: Given that the article heavily focuses on phosphorus levels and their correlation with other biomarkers such as CRP and IL-6, could the authors comment on the potential fluctuations in phosphorus levels during lymphodepletion in the patients?
Response: Thank you for this suggestion. We have added lines 230-233 to address this “Phosphorus levels remained stable during the lymphodepletion period before CAR-T infusion, suggesting that the observed decline in phosphorus levels was a direct effect of the infused CAR-T cells (figure 2).”
Comment: Based on the hypothesis, It is interesting to know that the levels of phosphorus can be corelated with CRS and its potential use as a possible biomarker. What are possible pathways that are involved in this, did the authors look into the mechanism by performing NGS on the samples ?
Response: Thank you for raising this point, we have further refined the explanation of the mechanism of the drop in lines 233-236. “ The mechanism behind the drop seen in phosphate levels with both CRS and ICANS appears to be multifactorial, as previous studies have shown CAR-T induced overaction from the immune system can both increase metabolic systemic ATP consumption as well as induced renal phosphate wasting. (14)(17)” We ourselves have not elucidated any pathways, but to our knowledge these two studies have given the most insight into the mechanism.
Comment: Is there a direct co-relation of phosphorus and CAR-T therapy ?
Response: To our knowledge, only as described above in lines 233-236. We have added figure 4 to try and illustrate the system inflammatory cascade and renal dysfunction with the resulting phosphorous drop.
Comment: The article primarily focuses on phosphorus levels in grade-2 Cytokine Release Syndrome (CRS). Did the authors also investigate other grades of CRS? If so, can any conclusions be drawn from the available data ?
Response: Unfortunately our limited data size precluded any further analysis of higher grades CRS and ICANS which we added acknowledgement in lines 281-283 “Our sample size limited the analysis of these scores in predicting progression to severe CRS and ICANS.”
Comment: Did the patients undergo any other treatment when IL-6 expression was higher such as the use of Tocilizumab or other IL-6 inhibitors for treatment.
Response: Thank you for this point, we’ve added in lines 77-80 “IL-6 inhibitors, IL-1 inhibitors, and steroids were administered per institutional protocol independent of investigated inflammatory marker values including IL-6 values.”
Comment: Can the authors include a schematic depiction of summary of the article.
Respoonse: Thank you for the suggestion! We've added Figure 4 to provide a clearer summary.
Reviewer 2 Report
Comments and Suggestions for Authors
ThThe paper presents an innovative approach to enhancing the m-EASIX score by incorporating phosphorous and IL6, which appears promising for the improved detection of ICANS and CRS progression. However, the clarity and effectiveness of the figures and tables could be further improved. Specifically, Figure 2 seems overly scattered and could benefit significantly from the addition of a schematic diagram. This diagram should clearly illustrate the pathway from antigen recognition to the activation of T cells or CAR-T cells, and the subsequent cytokine release, focusing on IL-6, IFN-γ, and TNF-α. It should also detail the roles of key cellular contributors like CAR-T cells, monocytes/macrophages, and endothelial cells in CRS. Moreover, ensuring all figures and tables, especially Table 3, are presented in high-resolution vector formats will enhance legibility and overall visual presentation. Lastly, refining Figures 2 and 3 to minimize clutter and improve data presentation would aid in reader comprehension and appreciation of the study's finding
Comments on the Quality of English Languagegood
Author Response
Comment: “The clarity and effectiveness of the figures and tables could be further improved. Specifically, Figure 2 seems overly scattered and could benefit significantly from the addition of a schematic diagram. This diagram should clearly illustrate the pathway from antigen recognition to the activation of T cells or CAR-T cells, and the subsequent cytokine release, focusing on IL-6, IFN-γ, and TNF-α. It should also detail the roles of key cellular contributors like CAR-T cells, monocytes/macrophages, and endothelial cells in CRS.”
Response: Thank you for your comments. In response, we created figure 4 to provide a schematic diagram describing the process of toxicities developing from time of CAR-T antigen recognition to key cytokine drivers and cellular responses of the systemic inflammatory response leading to CRS and ICANS.
Comment: Moreover, ensuring all figures and tables, especially Table 3, are presented in high-resolution vector formats will enhance legibility and overall visual presentation. Lastly, refining Figures 2 and 3 to minimize clutter and improve data presentation would aid in reader comprehension and appreciation of the study's finding.”
Agreed, we have also modified the graphs and figures mentioned to appear less cluttered.
Reviewer 3 Report
Comments and Suggestions for Authors
In the present manuscript the author try to establish a novel concept towards management of CAR-T related toxicity. Overall manuscript look good but certain concern need to resolve before further process. Such as
Authors, need to provide the data regarding patient history before treatment starts.
Further any comorbidity observed or not .
Any preclinical data of this type of therapy or any observation regarding toxicity.
Author, presented here the liquid cancer based therapy where as solid cancer data required to compare.
Furher need to provide any solution regarding overcome the toxicity.
Author Response
Thank you for your comments; we addressed them below as follows.
- Authors, need to provide the data regarding patient history before treatment starts.
We have included Table 1 (lines 127–128) to provide a detailed overview of relevant patient demographics and medical history, which are key considerations in determining CAR-T treatment eligibility in accordance with institutional protocols and standard of care guidelines.
- State if any comorbidity were present?
We utilized the ECOG score as a standardized and comprehensive measure of overall functional status and as a surrogate for comorbidities, in alignment with previous studies that have examined predictors of CAR-T-related toxicities.https://doi.org/10.1182/bloodadvances.2020003885
https://doi.org/10.1007/s00277-024-05617-y
https://doi.org/10.3390/cancers15225443
- Provide preclinical data of this type of therapy or any observation regarding toxicity.
In lines 58-61, we have included and expanded on the in vitro findings. “The study by Tang et al first demonstrated in vitro that increased consumption of phosphorous was associated with increased levels of CAR-T immune system activation to meet higher levels of metabolic demand induced by overaction of the immune system(14).”
- Discuss comparison to solid cancers, if relevant.
Based on our literature review, to the best of our knowledge, no CAR-T products are currently approved for use in solid cancers. While we identified some CAR-T products in Phase I/II trials for select solid tumors (listed below), the CAR-T products we investigated target B-cell maturation antigen, whereas those being explored for solid malignancies have different antigen targets. Given these differences and the lack of clinical data, a direct comparison of their toxicity profiles would be challenging.
https://doi.org/10.1186/s12935-024-03315-3
- Please discuss possible solutions regarding overcoming the toxicity.
We have added in lines 280-283 “These variations of the m-EASIX score aim to help predict CAR-T-related toxicities and guide clinical decision-making by identifying patients who may benefit from early interventions, such as steroids, IL-6 inhibitors, and IL-1 inhibitors, to prevent CAR-T toxicity onset and progression.”
Round 2
Reviewer 3 Report
Comments and Suggestions for Authors
Thanks for sharing the response and noted.